# Retinal Microvascular Abnormalities Predict Clinical Outcomes in Patients with Heart Failure

**DOI:** 10.3390/diagnostics12092078

**Published:** 2022-08-27

**Authors:** Shaohua Guo, Songtao Yin, Wenhua Song, Gary Tse, Juping Liu, Kaiwen Hei, Kangyin Chen, Long Su, Tong Liu

**Affiliations:** 1Tianjin Key Laboratory of Ionic-Molecular Function of Cardiovascular Disease, Department of Cardiology, Tianjin Institute of Cardiology, Second Hospital of Tianjin Medical University, Tianjin 300211, China; 2Department of Ophthalmology, Second Hospital of Tianjin Medical University, Tianjin 300211, China; 3Pears Building, Kent and Medway Medical School, University of Kent, Canterbury CT2 7FS, UK; 4Tianjin Key Laboratory of Retinal Functions and Diseases, Eye Institute and School of Optometry, Tianjin Medical University Eye Hospital, Tianjin 300384, China

**Keywords:** heart failure, retinal vessel analysis, microcirculation, prognosis

## Abstract

Background: Narrower retinal arterioles and wider retinal venules have been associated with the incidence of heart failure (HF). However, whether they are predictive of the prognosis of heart failure (HF) is unclear. We aimed to explore the role of retinal vessel calibers in predicting long-term clinical outcomes of HF. Methods: This is a prospective, single-center, observational study that surveyed patients in a tertiary referral hospital for the treatment of HF. Retinal vessel caliber was graded using retinal photography. The primary endpoint was the composite endpoint of HF rehospitalization and mortality at 12 months. Results: There were 55 patients with chronic HF included in the final analysis. At 12 months, the cumulative incidence of the primary endpoint, HF rehospitalization, and mortality tended to be higher with the widening of the central retinal venular equivalent (CRVE) (*p* for non-linearity = 0.059) and was significantly increased when CRVE reached a cut-off value (283 μm) (*p* = 0.011) following adjustment for age, sex, etiology of HF, and diabetes. No association between the central retinal arteriolar equivalent (CRAE) and arteriolar-to-venular caliber ratio (AVR) was found with the clinical outcome in both univariable and multivariable Cox regression. CRAE, CRVE, and AVR had no relationship with the concentration of the N-terminal pro-B-type natriuretic peptide. In addition, CRVE was not associated with cardiac diastolic and systolic function. Conclusions: When the retinal venular caliber widens to a certain point, the composite incidence of HF rehospitalization and mortality significantly increase, suggesting retinal vessel caliber imaging may provide insight into the development of HF.

## 1. Introduction

Heart failure (HF) has been recognized as one of most common causes of cardiovascular morbidity and mortality. Currently, there are 64.3 million HF patients worldwide, and the prevalence in the general adult population of developed countries is 1–2% and increases with age [1]. Despite significant progress in pharmacological and device therapy for HF, its long-term prognosis remains poor and its pathophysiology is incompletely understood.

Endothelial dysfunction is a pathophysiological mechanism in heart failure and represents a sensitive marker for disease severity and outcome prognosis [2]. Reduced blood flow and impairment of microcirculation play an important role in the systemic effects of heart failure on different organs [3]. Recently, retinal vessel analysis, a non-invasive and radiation-free method, has been widely explored for its potential application for HF prediction. Nagele et al. found an association between retinal microvascular abnormalities and the prevalence of HF of either ischaemic or non-ischaemic etiology, with a suggestion of a slightly downregulated function of retinal microcirculation [4]. In addition, data from the Atherosclerosis Risk in Communities (ARIC) Study suggested that wider retinal venular calibers have a predictive value for incident HF, which remained statistically significant following adjustment for traditional risk factors [5]. However, the prognostic value of retinal microcirculation for heart failure patients has been not defined. Therefore, we aimed to assess the association between retinal arteriolar and venule caliber with 12-month HF rehospitalization and mortality in heart failure patients.

## 2. Materials and Methods

### 2.1. Study Population

This was a prospective cohort clinical trial that evaluated the value of retinal vascular diameters in predicting the prognosis of chronic heart failure patients (ClinicalTrials.gov Identifier: ChiCTR1800018244). The study was conducted according to the tenets of the Declaration of Helsinki and the principles of good clinical practice guidelines [6]. The study protocol was approved by the local ethics committee of the Second Hospital of Tianjin Medical University (KY2018K063). All participants provided written informed consent.

Participants were recruited from October 2018 to June 2020 at the Second Hospital of Tianjin Medical University. The inclusion criteria for chronic heart failure accorded with the European Guidelines for the diagnosis and treatment of acute and chronic heart failure [7]: their left ventricular ejection fraction (LVEF) was lower than 40% and they had symptoms and/or signs of chronic heart failure (CHF) or their LVEF was ≥40% and their N-terminal pro-B-type natriuretic peptide (NT-proBNP) value was higher than 125 pg/mL with evidence of relevant structural heart disease such as left ventricular hypertrophy or atrial enlargement and/or diastolic dysfunction. Exclusion criteria included the following: (1) combined with other serious diseases, life expectancy was less than one year; (2) pregnancy or breastfeeding; (3) photosensitive epilepsy; (4) glaucoma, and (5) other significant eye pathologies and procedures such as blindness, inability to fixate (for instance, nystagmus), progressive diabetic retinopathy, or prior retinal laser coagulation.

### 2.2. Retinal Vessel Caliber Measurements

A 45-degree retinal photograph of a randomly selected eye of each participant was taken using a fundus camera (AFC-330; NIDEK Co., Ltd., Aichi, Japan) within 20–30 min after the application of two drops of 0.5% tropicamide solution. The photograph was centered on the region of the optic disc and was taken using an autofocus camera. At least 6 segments of retinal arteries and 6 segments of retinal vein diameters in an area of 0.5–1 optic disc diameter distant from the optic disc were added with the calculation of the central retinal artery and vein equivalent (CRAE and CRVE) using IVAN software [8]. With the use of formulas presented by Parr and Spears [9,10] and by Hubbard et al. [11], these measurements were combined into 3 summary indices, representing the average retinal arteriolar diameters (CRAE), retinal venular diameters (CRVE), and arteriole-to-venule ratio (AVR = CRAE/CRVE) (Figure 1).

Each image was analyzed by well-trained researchers (S.G., S.Y., and K.H.), supervised and in the case of discrepancy, finally validated by the most experienced scientists (J.L. and L.S.) of the group to assure uniform and correct data acquisition.

### 2.3. Clinical, Biochemical, and Echocardiographic Data

Baseline data concerning demographic and clinical variables involving age, sex, hospital stay, etiology of HF, smoking history, comorbidities (such as hypertension, diabetes, coronary revascularization history, stroke, and atrial fibrillation), blood pressure, heart rate, biochemical results (in particular, NT-proBNP), and discharge medication were collected. Participants underwent transthoracic echocardiography performed by trained sonographers using Philips IE33 machines (Philips Ultrasound, Bothell, WA, USA). LVEF was measured based on the modified biplane Simpson’s method. The measurements of the left atrial diameter (LAD), interventricular septum thickness, left ventricular end-diastolic diameter, and left ventricular end-systolic diameter were from parasternal long-axis view. Early septal mitral annular velocities (e’) were measured using tissue Doppler imaging. Doppler measurements of mitral inflow were made from two-dimensional views.

### 2.4. Follow-Up and Outcomes

All patients with HF continued the standardized treatment for HF after discharge. Patients were followed up by clinical visits or telephone calls for 12 months. The primary endpoint was the composite endpoint of all-cause mortality and HF rehospitalization at 12 months. The follow-up time was calculated from discharge to all-cause mortality, first readmission, or termination of the study.

### 2.5. Statistical Analysis

Patients were divided into two groups based on the occurrence of the primary endpoint. Continuous variables were reported as the mean ± SD or median and interquartile range. Student’s-test was used for data with a normal distribution, and the Mann–Whitney test was used for data with a nonnormal distribution. Categorical variables were expressed as numbers and percentages and were compared using the Chi-square test or Fisher’s exact test. CRVE, CRAE, and AVR were analyzed as continuous variables.

Cox proportional hazards regression was used to calculate the hazard ratios (HR) among all patents. Multivariable models investigating the association between CRVE or CRAE and the primary endpoint were created, adjusting for age, sex, etiology of HF, and diabetes. Non-linearity was observed in the relationship between CRVE and the primary endpoints in restricted cubic spline regression analysis. According to the result of restricted cubic spline regression analysis, a cut-off value of CRVE was found to define the patients with higher risk. Log-rank tests for the Kaplan–Meier survival curves were performed according to subgroups divided by this cut-off value. Correlations between levels of CRAE, CRVE, AVR, and N-terminal pro-B-type natriuretic peptide (NT-proBNP) were assessed using Pearson’s r correlation. Linear and logistic regressions were applied to investigate the associations between echocardiographic structural/functional characteristics and CRVE at baseline.

All data were analyzed using SPSS statistical software (SPSS 25.0, SPSS Inc., Chicago, IL, USA) and R programming (version 4.1.2; R Foundation for Statistical Computing, Vienna, Austria). A *p*-value < 0.05 was considered statistically significant.

## 3. Results

Baseline Characteristics

A total of 70 patients met the inclusion criteria. Of these, 15 patients were excluded because of a lack of available retinal photographs to analyze, intolerant of waiting for fundus photography or lost contact. Therefore, 55 patients with HF (mean age 65.6 ± 12.7 years; 79.9% men) were included in the final analysis (Figure 2).

At 12 months, the primary endpoint (HF rehospitalization and mortality) occurred in 22 (40%) patients, of whom 4 (7.3%) died and 18 (32.7%) were rehospitalized for HF. Compared to patients who did not have the primary endpoints, patients who developed the primary endpoints were more likely to have a revascularization history. No difference was found in age, sex, etiology of HF, medical history, laboratory parameters, and the prescribed medicines between the two groups. Otherwise, no difference was found between the two groups in CRAE, CRVE, and AVR (Table 1).

For each standard deviation change, CRAE, CRVE, and AVR were not associated with the risk of HF rehospitalization and mortality. After adjusting for age, sex, etiology of HF, and diabetes, the association remained insignificant (Table 2). However, the CRVE showed a trend of a higher risk of rehospitalization and mortality (*p*-value for overall trend = 0.129; *p*-value for non-linearity = 0.059), but no significant non-linear association was detected for CRAE (*p*-value for non-linearity = 0.229) and AVR (*p*-value for non-linearity = 0.236) (Figure 3). As shown in Figure 3B, a cut-off value (283 μm) of CRVE was found to have a prognostic value. Based on a comparison of the baseline characteristics (Appendix A), patients with CRVE ≥ 283 μm tended to be younger than patients with CRVE ≥ 283 μm. HF patients with CRVE ≥ 283 μm at baseline had a 2.7 times higher risk of HF rehospitalization and mortality than HF patients with CRVE < 283 μm (*p* = 0.032, Figure 4). Wider CRVE (≥283 μm) (hazard ratio [HR] 3.82, 95% confidence interval [CI] 1.36 to 10.72; *p* = 0.011), adjusted for age, sex, etiology of HF, and diabetes, was still significantly associated with HF rehospitalization and mortality (Figure 5). CRAE, CRVE, and AVR did not correlate with the NT-proBNP levels (Figure 6).

Furthermore, widening CRVE was not associated with cardiac structural characteristics such as left atria dimensions, left ventricular end-diastolic dimensions, left ventricular end-systolic dimensions, septal wall thickness, and posterior wall thickness and cardiac structural characteristics such as left ventricular ejection fraction, septal E/e’, and pulmonary arterial hypertension (Figure 7).

## 4. Discussion

In this study, we demonstrated that the caliber of retinal venules (CRVE) and narrower retinal arterioles (CRAE) were not associated with 12-month HF rehospitalization and mortality in HF patients, but widening of CRVE above the cut-off value of 283 μm was significantly associated with 12-month HF rehospitalization and mortality. To the best of our knowledge, this is the first study exploring the role of the retinal vascular caliber in the prognosis of HF patients.

In recent years, a growing body of evidence has suggested an association of various risk factors of HF with retinal microvascular signs, such as older age [12], hypertension [13], diabetes mellitus [14], coronary disease [15,16], stroke [17,18], and atrial fibrillation [19]. Retinal vessels are approximately the same size as the coronary and cerebral microvasculature and share embryologic, anatomic, and physiologic similarity. Therefore, retinal microvascular dysfunction may reflect subclinical pathological changes prior to the development of cardiovascular disease and diagnosis. The Atherosclerosis Risk in Communities (ARIC) in 2016 reported that narrower retinal arterioles and wider retinal venules conferred a long-term risk of coronary heart disease in females and ischemic stroke in both sexes [18], as confirmed by the meta-analysis conducted in 2020 [20]. There is also evidence demonstrating an increased risk of mortality in individuals with both narrower retinal arteriolar caliber and wider retinal venular caliber [18,21]. All evidence compelled investigators to further investigate the association between retinal vessel calibers and HF.

Currently, few studies have explored the role of retinal vessel calibers in HF; however, there are discrepancies among studies. In 2015, a cross-sectional study involving 1680 participants found a significant association between a wider caliber of retinal venules and the prevalence of heart failure [22]. Meanwhile, the ARIC study [18] published in 2016 did not find a significant association between retinal vessel caliber and heart failure, findings that are supported by a case-control study in 2018 [4]. Surprisingly, later evidence from the ARIC study group published in 2019 found that following adjustment for traditional risk factors, wider CRVE remained the statistically significant predictor of incident HF after a long-term follow up [5]. These findings therefore demonstrate that changes in retinal vessel caliber may only demonstrate a predictive value over longer periods. No association of wider CRVE and narrower CRAE with the prognosis of HF was found in our study, even after multivariable adjustment.

Despite all of these explorations of the relationship between retinal vessels and cardiovascular disease, there is no study on the role of retinal vessel calibers in the prognosis of HF, which may mainly be due to HF patients being usually frail and intolerant to conducting a time-consuming examination such as fundus photography. Our study builds upon these findings, showing that CRVE presented a trend of a higher risk of rehospitalization and mortality in non-linear analysis (*p* = 0.059), with a significant tendency when the CRVE widens beyond a certain value (*p* = 0.032). However, CRAE narrowing was not linearly or non-linearly associated with an increased risk of HF rehospitalization and mortality. This echoes the findings from ARIC in 2019 [5], demonstrating that CRVE has the potential value of predicting not only the incidence but also the prognosis of HF. A study with a large sample size is necessary to further explore the non-linear relationship between CRVE and the prognosis of HF.

In addition, no relation was found between the caliber of retinal veins and NT-proBNP, a well-recognized prognostic marker and indicator of elevated ventricular filling pressure among patients regardless of ejection fraction [23,24], indicating that a wider CRVE may serve as a supplementary not a replacement to estimate prognosis in HF patients.

Regarding cardiac structure and function, the ARIC study reported in 2019 that among individuals free of incident HF at baseline, a wider CRVE and narrower CRAE were significantly associated with an increased left ventricular size, a higher incidence of left ventricular hypertrophy, and greater abnormalities in diastolic and systolic functions, which notably differed by sex [5]. Our study found no relationship between CRVE and the cardiac structural and functional parameters among HF patients, suggesting that retinal microvascular abnormalities may subtend a continuum of left ventricular remodeling at a very early time before overt HF, but this cannot explain the different magnitudes of left ventricular remodeling.

Several limitations should be noted. First, the present study is a non-randomized, single-center analysis and is limited by its observational nature. However, we performed multivariable analysis, trying to balance the bias and mimic a randomized setting. Secondly, single-center experience with a limited sample size affects its wide application. Current trends observed in this study should be confirmed by future, larger prospective, adequately powered population studies. Third, due to the limited population size, subgroup analyses for HF subtypes, sex, and other factors were not performed and should be conducted in future larger scale studies.

## 5. Conclusions

In summary, the current study explored the association of retinal vessel calibers with clinical outcomes in HF patients. Wider retinal venous caliber was significantly associated with 12-month HF rehospitalization and mortality in HF patients.

## Figures and Tables

**Figure 1 diagnostics-12-02078-f001:**
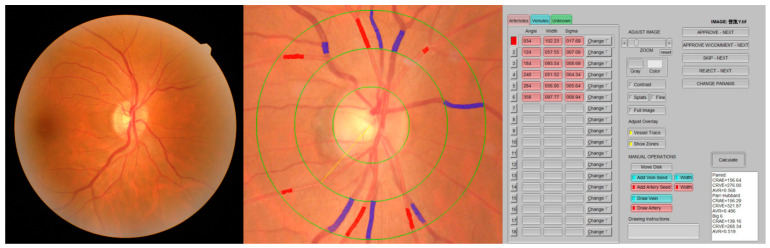
Retinal vessel caliber measurements using IVAN software.

**Figure 2 diagnostics-12-02078-f002:**
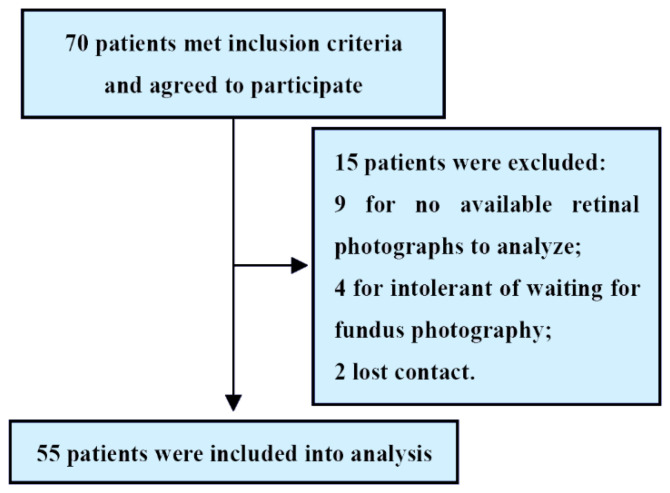
Study flowchart.

**Figure 3 diagnostics-12-02078-f003:**
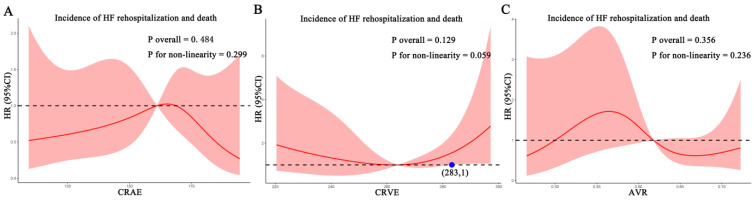
Association of retinal vessel caliber with risk for HF rehospitalization and mortality in HF patients. (**A**) for CRAE; (**B**) for CRVE, (**C**) for AVR. AVR, arteriolar-to-venular caliber ratio; CI, confidence interval; CRAE, central retinal arteriolar equivalent; CRVE, central retinal venular equivalent; HF, heart failure. HR, hazard ratio.

**Figure 4 diagnostics-12-02078-f004:**
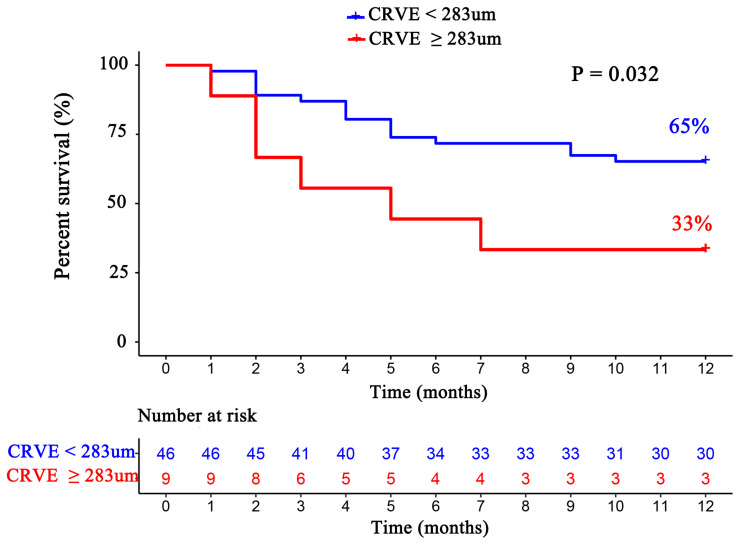
The Kaplan–Meier survival curves according to subgroups divided by CRVE (CRVE ≥ 283 μm and CRVE < 283 μm). CRVE, central retinal venular equivalent.

**Figure 5 diagnostics-12-02078-f005:**
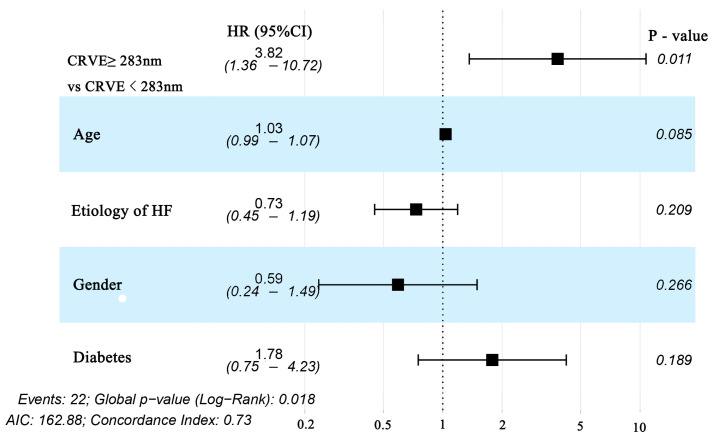
Forest plot of subgroups divided by CRVE (CRVE ≥ 283 μm and CRVE < 283 μm) for Cox regression with multivariable adjustment for age, sex, etiology of HF, and diabetes for the prediction of the primary endpoint (HF rehospitalization and mortality at 12-month follow-up) in patients with heart failure. CRVE, central retinal venular equivalent; HF, heart failure.

**Figure 6 diagnostics-12-02078-f006:**
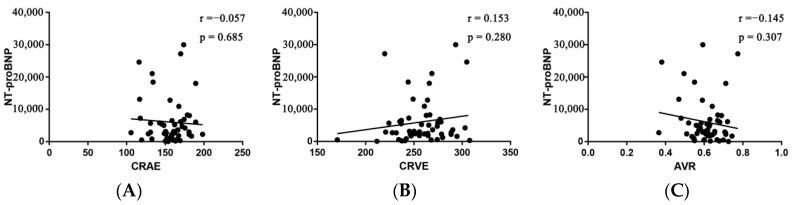
Pearson’s correlation between NT-proBNP and CRAE (**A**), CRVE (**B**), and AVR (**C**). AVR, arteriolar-to-venular caliber ratio; CRAE, central retinal arteriolar equivalent; CRVE, central retinal venular equivalent; NT-proBNP, N-terminal probrain natriuretic peptide.

**Figure 7 diagnostics-12-02078-f007:**
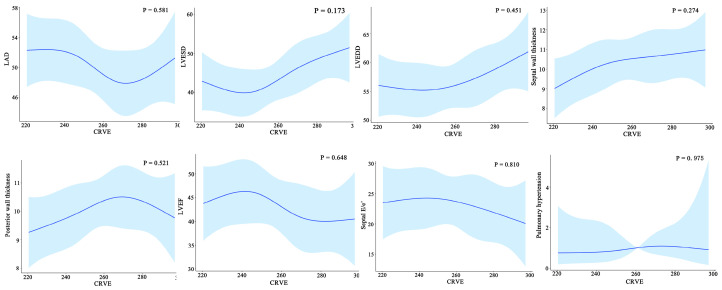
Relationship between the cardiac structure/function and CRVE of HF patients. CRAE, central retinal arteriolar equivalent; HF, heart failure. LAD, left atrial diameter; LVESD, left ventricular end-systolic diameter; LVEDD, left ventricular end-diastolic diameter.

**Table 1 diagnostics-12-02078-t001:** Demographic and baseline characteristics of HF patients with endpoints and without endpoints.

Variables	HF Patients with Endpoints (*n* = 22)	HF Patients without Endpoints (*n* = 33)	*p* Value
Age (years)	68.86 ± 11.97	63.45 ± 12.91	0.123
Male (%)	8 (24.2)	25 (75.8)	0.332
Hospital stay (days)	9.41 ± 5.07	9.09 ± 2.94	0.770
Systolic BP (mmHg)	134.95 ± 22.47	142.55 ± 29.06	0.305
Diastolic BP (mmHg)	84.32 ± 13.32	89.00 ± 18.68	0.315
Heart rate (b.p.m.)	78 (74–94)	89 (75–103)	0.080
Etiology of HF			
Ischemic (%)	13 (59.1)	10 (30.3)	0.106
Dilated (%)	2 (9.1)	5 (15.2)	
Other (%)	7 (31.8)	18 (54.5)	
NYHA class			
I (%)	0 (0)	0 (0)	0.823
II (%)	5 (22.7)	8 (24.2)	
III (%)	13 (59.1)	21 (63.6)	
IV (%)	4 (18.2)	4 (12.1)	
Medical history			
Hypertension (%)	14 (63.6)	23 (69.7)	0.639
Diabetes mellitus (%)	13 (59.1)	13 (39.4)	0.152
Myocardial farction (%)	7 (31.8)	6 (18.2)	0.244
Revascularization (%)	10 (45.5)	5 (15.2)	0.013
Atrial fibrillation (%)	11 (50.0)	12 (36.4)	0.315
Dyslipidemia	6 (27.3)	10 (30.3)	0.808
Stroke (%)	7 (27.3)	8 (24.2)	0.800
Smoker (%)	4 (18.2)	13 (39.4)	0.095
Echocardiographic indicators			
LVEF (%)	40.0 (32.0–59.0)	40.0 (28.0–58.0)	0.585
LAD (mm)	46.8 (42.8–61.8)	52.3 (47.0–54.8)	0.495
LVESD (mm)	49.7 (30.7–53.8)	45.9 (30.5–55.0)	0.931
LVEDD (mm)	58.5 (49.6–62.9)	58.1 (49.1–62.8)	0.630
Septal wall thickness (mm)	9.7 (9.2–11.4)	9.5 (8.1–13.0)	0.863
Posterior wall thickness (mm)	9.5 (8.5–10.7)	9.9 (7.7–12.1)	0.730
Septal E/e’	21.4 (15.7–27.9)	20.7 (14.1–27.4)	0.519
TRPG	38.40 ± 13.45	33.80 ± 12.73	0.442
Pulmonary hypertension (%)	6 (28.6)	10 (32.3)	0.777
Laboratory Tests			
Triglycerides (mM)	1.16 ± 0.54	1.36 ± 1.03	0.440
Total cholesterol (mM)	4.27 ± 1.32	4.32 ± 1.27	0.891
LDL (mM)	2.74 ± 1.12	2.79 ± 0.97	0.870
HDL (mM)	1.06 ± 0.32	1.5 ± 0.31	0.912
Fasting glucose (mM)	6.18 (5.26–6.68)	6.57 (4.92–9.53)	0.579
Creatinine (μM)	108.33 ± 51.22	100.47 ± 46.43	0.558
eGFR (mL/min/BSA)	68.57 (48.56–88.62)	78.03 (54.86–96.66)	0.439
NT-proBNP (ng/L)	3621.00 (843.00–8266.00)	2791.10 (1850.00–6891.00)	0.638
Troponin T (ng/mL)	0.01 (0.00–0.26)	0.01 (0.00–0.05)	0.497
Haemoglobin (g/L)	129.00 ± 27.30	134.58 ± 20.46	0.391
Medication			
ACEI/ARB/ARNI (%)	18 (81.8)	25 (75.8)	0.594
Beta-blocker (%)	16 (72.7)	27 (81.8)	0.424
Loop diuretic (%)	16 (72.7)	21 (63.6)	0.481
Statin	13 (59.1)	23 (69.7)	0.418
Mineralocorticoid receptor antagonist (%)	13 (59.1)	23 (69.7)	0.418
Calciμm channel blocker (%)	1 (14.8)	13 (27.1)	0.468
Oral anticoagulation (%)	8 (36.4)	11 (33.3)	0.817
Retinal vessel calibers			
CRAE (μm)	159.7 (140.6–169.5)	158.9 (146.8–172.1)	0.918
CRVE (μm)	265.2 (236.4–285.9)	260.8 (246.5–271.4)	0.837
AVR	0.59 (0.55–0.67)	0.63 (0.56–0.68)	0.525

ACEI, angiotensin converting enzyme inhibitor; ARB, angiotensin receptor blocker; ARNI, angiotensin receptor-neprilysin inhibitors; AVR, arteriolar-to-venular caliber ratio; BP, blood pressure; b.p.m, beat per minute; CRAE, central retinal arteriolar equivalent; CRVE, central retinal venular equivalent; HF, heart failure; LAD, left atrial diameter; LVEF, left ventricular ejection fraction; LVESD, left ventricular end-systolic diameter; LVEDD, left ventricular end-diastolic diameter; E/e’, early transmitral inflow to early mitral relaxation velocity ratio; LDL, low-density lipoprotein; HDL, high-density lipoprotein; eGFR, estimated glomerular filtration rate; NT-proBNP, N-terminal probrain natriuretic peptide; NYHA, New York Heart Association; TRPG, tricuspid regurgitation peak gradient.

**Table 2 diagnostics-12-02078-t002:** Associations of retinal vessel calibers (per one standard deviation change) with endpoints in HF participants at baseline.

Model	Events	Per SD Hazard Ratio (95% CI)		
		CRAE	CRVE	AVR
Unadjusted model	22 (40%)	0.97 (0.65–1.45) *p* = 0.882	1.03 (0.97–1.60) *p* = 0.901	0.98 (0.65–1.46) *p* = 0.906
Model adjusted for age, sex, etiology of HF, and diabetes	22 (40%)	1.11 (0.72–1.71) *p* = 0.642	1.13 (0.71–1.81) *p* = 0.608	1.07 (0.66–1.74) *p*= 0.788

AVR, arteriolar-to-venular caliber ratio; CI, confidence interval; CRAE, central retinal arteriolar equivalent; CRVE, central retinal venular equivalent; HF, heart failure; SD, standard deviation.

## Data Availability

The data presented in this study are available on request from the corresponding author. The data are not publicly available due to patient privacy.

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
