# Peer review of "Retinal Microvascular Abnormalities Predict Clinical Outcomes in Patients with Heart Failure"

_diagnostics, 2022, doi:10.3390/diagnostics12092078_

Round 1

Reviewer 1 Report

In this study Guo and colleagues performed a prospective observational study in heart failure patients with the intent of exploring possible associations between clinical outcomes of the disease (rehospitalization and mortality rates) and parameters of the retinal microvasculature. This study is relevant for the building current knowledge on the relevance of retinal assessment for the prognosis of cardiovascular disease. The study seems to have been well-conducted and the manuscript is well-written. I have several major and minor issues regarding the manuscript:

Major

11. In order to further support the need to conduct this study and its overall relevance, I would suggest that the authors include a sentence/paragraph describing the morphofunctional similarities between the retinal and coronary/cerebral microvasculature;

22. Just as the authors demonstrated, previous papers have shown a poor/moderate correlation between echocardiography heart failure parameters and retinal microvascular parameters. However, did the authors have access to any parameters directly/indirectly relating to vascular function (coronary perfusion, skin/muscle perfusion secondary to reactive hyperemia, homocysteine, D-dimer, others) of patients, considering that several had other diseases that increased vascular risk? If so, other correlations could be tested;

33. Did the authors only assess vascular diameters? Did they also calculate vascular tortuosity? Also, was arteriovenous nicking and/or edema present in any patient?

44. In my opinion the discussion on the differences between the present study and the ARIC study can be expanded;

Minor

11. An explanation should be provided as to how the authors determined their sample size;

22. In which hospital services/specialties did the authors recruit patients?

33. Please define LVEF and CHF the first time these terms appear in the text;

44. Please add a reference regarding the statement on following the declaration of Helsinki;

55. I encourage the authors to move their supplementary material into the manuscript itself since it facilitates reading;

66. Why did the authors only present one p-value for the etiology of HF and NYHA class, considering there are different classes within each category?

77. Several spelling mistakes and typos are present along the text, for example:

a.       line (L) 22: it should be “incidence”

b.      L53 and 214: it should be “prevalence”

c.       Table 1: it should be “fraction”

d.      L171: it should be “correlate”

   8. On line 78 please specify the diseases where patients had inability to fixate; in the same line I suggest to add “and procedures” after “eye pathologies”;

99. I suggest that the authors use the term “sex” instead of “gender” throughout the text;

110) I suggest that the authors use the term “mortality” instead of “death”;

111) Please include the symbol “μ” instead of “u” throughout the manuscript (for example on table 1);

112) Please improve the caption of table 1;

113) Please increase the size of figure 2;

Author Response

We are very grateful to Reviewers for reviewing the paper so carefully. These comments are very valuable and helpful for revising and improving our paper. We have carefully considered the suggestion of Reviewers and made some changes according to the reviewers’ recommendations. {Liew, 2007 #30017}All corrections in the paper are marked in yellow font. In specific we provide the following point-by-point comments:

Major

  1. In order to further support the need to conduct this study and its overall relevance, I would suggest that the authors include a sentence/paragraph describing the morphofunctional similarities between the retinal and coronary/cerebral microvasculature;

Thanks very much for your helpful comments, we agree with your opinion. The sentence “Retinal vessels are approximately the same size as the coronary and cerebral microvasculature, share embryologic, anatomic, and physiologic similarity” were added on page 13 line13-14. 

  1. Just as the authors demonstrated, previous papers have shown a poor/moderate correlation between echocardiography heart failure parameters and retinal microvascular parameters. However, did the authors have access to any parameters directly/indirectly relating to vascular function (coronary perfusion, skin/muscle perfusion secondary to reactive hyperemia, homocysteine, D-dimer, others) of patients, considering that several had other diseases that increased vascular risk? If so, other correlations could be tested;

Thanks for your advice. The parameters of coronary perfusion, skin/muscle perfusion secondary to reactive hyperemia, and homocysteine were not available in our database. We have performed the analysis to explore the correlation of retinal vessel calibers and D-dimer, no statistical significance was found.

3. Did the authors only assess vascular diameters? Did they also calculate vascular tortuosity? Also, was arteriovenous nicking and/or edema present in any patient?

Unfortunately, we only assessed vascular diameters. 

  1. In my opinion the discussion on the differences between the present study and the ARIC study can be expanded;

Thanks for your advice. For now, only two paper focusing on the correlation of retinal microvasculature and HF were published, and we tried to expand the discussion on the differences as following: “While, no association of wider CRVE and narrower CRAE with prognosis of HF was found in our study, even after multivariable adjustment ” on page 14 line 5-6 and “This echoes the findings from ARIC in 2019, demonstrating that CRVE have the potential value of  predicting not only the incidence but also the prognosis of HF. Large sample size study is necessary to further explore the non-linear relationship with CRVE and prognosis of HF ” on page 14 line 13-15.

Minor

  1. An explanation should be provided as to how the authors determined their sample size;

As HF patients being usually frail and intolerant to conducting a time-consuming examination like fundus photography, it is really hard to recruit HF patients. Therefore the sample of this prospective study was severely limited, and we tried to recruit patients as many as possible without a target sample size.

  1. In which hospital services/specialties did the authors recruit patients?

We only recruit patients in the Second Hospital of Tianjin Medical University.

  1. Please define LVEF and CHF the first time these terms appear in the text;

Thanks for your helpful comments. The definition of LVEF and CHF was added on page 5 line 1 and page 3 line 28, respectively.

  1. Please add a reference regarding the statement on following the declaration of Helsinki;

The reference was added as 6 on page 3 line 23.

  1. I encourage the authors to move their supplementary material into the manuscript itself since it facilitates reading;

We agree with your opinion. The supplementary material have been moved into manuscript.

  1. Why did the authors only present one p-value for the etiology of HF and NYHA class, considering there are different classes within each category?

Considering the limited sample size, further analysis of difference between classes within category would not be meaningful.

  1. Several spelling mistakes and typos are present along the text, for example:

1)   line (L) 22: it should be “incidence”

2)   L53 and 214: it should be “prevalence”

3)   Table 1: it should be “fraction”

4)   L171: it should be “correlate”

Thanks for the your kind reminder, we have corrected the mistake. please see page 2 line 3, page 3 line 10, page 8 line 19, and page 13 line 23.

  1. On line 78 please specify the diseases where patients had inability to fixate; in the same line I suggest to add “and procedures”after “eye pathologies”;

We agree with your opinion. We enumerated nystagmus which can lead to inability to fixate on page 4 line 3, and the “and procedures”was also added after “eye pathologies” on page 4 line 2.

  1. I suggest that the authors use the term “sex” instead of “gender” throughout the text;

We agree with you on this point, and we have made corresponding changes in the revised manuscript.

  1. I suggest that the authors use the term “mortality” instead of “death”;

We agree with your opinion, and we have made the corresponding revisions.

  1. Please include the symbol “μ”instead of “u” throughout the manuscript (for example on table 1);

Thanks for the your kind reminder, all the “u” have been replaced with “μ”.

  1. Please improve the caption of table 1;

We have made the modification as see on page 6 line 16.

  1. Please increase the size of figure 2;

Figure 2 was modified according to your suggestion.

Reviewer 2 Report

This is a prospective, single-center, observational study, which aimed to assess the association between retinal arteriolar and venule caliber with 12-month HF rehospitalization and death in heart failure patients. The authors indicated the association of retinal vessel calibers with clinical outcomes in HF patients, in which wider retinal venous caliber was significantly associated with 12-month HF rehospitalization and death in HF patients. 

This reviewer considers that the present study might be clinically important. This reviewer has some comments as described below. 

Major comments:

1.     Table 1. The authors should include dyslipidemia of medical history and statin use in medication. 

2.     Also in Table 1, the authors should show tricuspid regurgitation peak gradient (TRPG) data combined with the prevalence of pulmonary hypertension. 

3.     Again in Table 1. The authors enrolled heart failure patients. Were the pulmonary arterial patients included in this population? Or, did the authors mean “pulmonary hypertension associated with left heart diseases”?  

4.     Also, the authors should include ECG data, especially left ventricular hypertension (with or without strain patten) and right ventricular overload (RV1 or RV1+SV5/6).

5.     The authors only showed patients characteristics compared between “HF patients with endpoints” and “HF patients without endpoints”. Comparison data of patients characteristics between “CRVE < 283um” and “CRVE ≥ 283 um” is also required.

Minor comment:

6.     In the Methods section in the Abstract. “a” might be necessary between “This is” and “prospective, single-centre, observational study”. 

Author Response

Dear Reviewer,

We are very grateful to Reviewers for reviewing the paper so carefully. These comments are very valuable and helpful for revising and improving our paper. We have carefully considered the suggestion of Reviewers and made some changes according to the reviewers’ recommendations. All corrections in the paper are marked in yellow font. In specific we provide the following point-by-point comments:

This is a prospective, single-center, observational study, which aimed to assess the association between retinal arteriolar and venule caliber with 12-month HF rehospitalization and death in heart failure patients. The authors indicated the association of retinal vessel calibers with clinical outcomes in HF patients, in which wider retinal venous caliber was significantly associated with 12-month HF rehospitalization and death in HF patients.

This reviewer considers that the present study might be clinically important. This reviewer has some comments as described below.

Major comments:

  1. Table 1. The authors should include dyslipidemia of medical history and statin use in medication.

Thanks for your helpful comments. The dyslipidemia of medical history and statin were added into Table 1.

  1. Also in Table 1, the authors should show tricuspid regurgitation peak gradient (TRPG) data combined with the prevalence of pulmonary hypertension.

The TRPG data has been added in Table 1.

  1. Again in Table 1. The authors enrolled heart failure patients. Were the pulmonary arterial patients included in this population? Or, did the authors mean “pulmonary hypertension associated with left heart diseases”?  

Thanks for this question. We recruited the heart failure patients, and some patients were combined with pulmonary hypertension. However, by etiological analysis, except for one patient with pulmonary heart disease, other patients with pulmonary hypertension were associated with left heart disease

  1. Also, the authors should include ECG data, especially left ventricular hypertension (with or without strain patten) and right ventricular overload (RV1 or RV1+SV5/6).

There is unfortunately no ECG data available.

  1. The authors only showed patients characteristics compared between “HF patients with endpoints” and “HF patients without endpoints”. Comparison data of patients characteristics between “CRVE < 283um” and “CRVE ≥ 283 um” is also required.

Comparison data of patients characteristics between “CRVE < 283μm” and “CRVE ≥ 283 μm”  was shown in Supplemental Table 1 as seen on page 9 line 5 to page 11 line 1.

Minor comment:

  1.     In the Methods section in the Abstract. “a” might be necessary between “This is” and “prospective, single-centre, observational study”.

Thanks for your advice, we have already added it.

Round 2

Reviewer 1 Report

The revised version of the manuscript by Guo and colleagues has been significantly improved, both in terms of content and structure, and can be accepted for publication. I thank the authors for taking into account this reviewer's suggestions.

I would like to suggest minor corrections/comments to this revised version:

1) Table 1. It should be "statin" and not "stain";

2) Page 3, Line 27. LVEF should be described here, the first time this abbreviation appears in the text, and not on Page 5, Line 1. 

3) Page 5, Line 1. Please provide the country of the manufacturer of the Philips IE33 equipment;

4) Page 5, Line 28. Please provide the country of the company that created the statistics softwares used in this study;

5) Since the supplemental figures are now in the manuscript, I suggest to remove the term "supplemental" from the caption;

6) Figure 3 should be enlarged (at least the front).

7) I agree with Reviewer 2, the authors should clarify whether they intended to write "Pulmonary arterial hypertension" or simply "Arterial hypertension" since the first is part of the second;

Author Response

Dear Reviewer,

Thanks very much for taking your time to review this manuscript. I really appreciate all  your comments and suggestions! Please find my itemized responses in below and my revisions/corrections in the re-submitted files. Thanks again!

1) Table 1. It should be "statin" and not "stain";

Sorry we meant for "statin". This part has been revised: please see Table 1 on page 8 and Supplemental Table 1 on page 11.

2) Page 3, Line 27. LVEF should be described here, the first time this abbreviation appears in the text, and not on Page 5, Line 1. 

Sorry for the mistake. LVEF has been redefined on Page 3, Line 27.

3) Page 5, Line 1. Please provide the country of the manufacturer of the Philips IE33 equipment;

The country of manufacturer have been added on page 5, Line 3.

4) Page 5, Line 28. Please provide the country of the company that created the statistics softwares used in this study;

The country of the company that created the statistics softwares used in this study were added on page 6 line 1-2.

5) Since the supplemental figures are now in the manuscript, I suggest to remove the term "supplemental" from the caption;

We agree with your point. The supplemental figures have been renumbered in the manuscript.

6) Figure 3 should be enlarged (at least the front).

Figure 3 (renamed as Figure 4) was modified according to your suggestion.

7) I agree with Reviewer 2, the authors should clarify whether they intended to write "Pulmonary arterial hypertension" or simply "Arterial hypertension" since the first is part of the second;

Thank you for your kind reminder.We meant for “pulmonary hypertension”, we have revised it in Table 1 on page 7 and Supplemental Table 1 on page 10.

Reviewer 2 Report

This prospective, single-center, observational study was well revised, but this reviewer still has a major comment as described below. 

Major comment:

1.     Table 1. The authors described pulmonary arterial patients; however, only a patient had pulmonary arterial hypertension. Pulmonary arterial hypertension is different from pulmonary hypertension associated with left heart diseases. The authors should clearly describe this issue in Table 1. Or, it should be “pulmonary hypertension”, not pulmonary arterial hypertension. 

Author Response

Dear Reviewer,

Thanks very much for taking your time to review this manuscript. I really appreciate all  your comments and suggestions! Please find my itemized responses in below and my revisions/corrections in the re-submitted files. Thanks again!

1) Table 1. The authors described pulmonary arterial patients; however, only a patient had pulmonary arterial hypertension. Pulmonary arterial hypertension is different from pulmonary hypertension associated with left heart diseases. The authors should clearly describe this issue in Table 1. Or, it should be “pulmonary hypertension”, not pulmonary arterial hypertension. 

Thank you for your kind reminder.We meant for “pulmonary hypertension”, we have revised it in Table 1 on page 7 and Supplemental Table 1 on page 10.